# Acylation of Anthocyanins and Their Applications in the Food Industry: Mechanisms and Recent Research Advances

**DOI:** 10.3390/foods11142166

**Published:** 2022-07-21

**Authors:** Xiu’er Luo, Ruoyong Wang, Jinhua Wang, Ying Li, Huainan Luo, Shi Chen, Xin’an Zeng, Zhong Han

**Affiliations:** 1School of Food Science and Engineering, South China University of Technology, Guangzhou 510641, China; ncuspyluoxiuer@163.com (X.L.); liying010062@163.com (Y.L.); luohuainan2021@scut.edu.cn (H.L.); feschen@mail.scut.edu.cn (S.C.); xazeng@scut.edu.cn (X.Z.); 2Air Force Medical Center PLA, Beijing 100142, China; wryafmcfmmu@163.com; 3Foshan Shunde Midea Washing Appliances MFG, Co., Ltd., Foshan 528300, China; wangjh12@midea.com; 4School of Food Science and Engineering, Foshan University, Foshan 528225, China; 5Guangdong Provincial Key Laboratory of Intelligent Food Manufacturing, Foshan University, Foshan 528225, China; 6Overseas Expertise Introduction Center for Discipline Innovation of Food Nutrition and Human Health (111 Center), Guangzhou 510641, China

**Keywords:** acylated anthocyanins, acylation, food colourants, functionalizing agents, indicators

## Abstract

Anthocyanins are extensively used as natural non-toxic compounds in the food industry due to their unique biological properties. However, the instability of anthocyanins greatly affects their industrial application. Studies related to acylated anthocyanins with higher stability and increased solubility in organic solvents have shown that the acylation of anthocyanins can improve the stability and fat solubility of anthocyanins. However, relevant developments in research regarding the mechanisms of acylation and applications of acylated anthocyanins are scarcely reviewed. This review aims to provide an overview of the mechanisms of acylation and the applications of acylated anthocyanins in the food industry. In the review, acylation methods, including biosynthesis, semi-biosynthesis, and chemical and enzymatic acylation, are elaborated, physicochemical properties and biological activities of acylated anthocyanins are highlighted, and their application as colourants, functionalizing agents, intelligent indicators, and novel packaging materials in the food industry are summarized. The limitations encountered in the preparation of acylated anthocyanins and future prospects, their applications are also presented. Acylated anthocyanins present potential alternatives to anthocyanins in the food industry due to their functions and advantages as compared with non-acylated analogues. It is hoped that this review will offer further information on the effective synthesis and encourage commercialization of acylated anthocyanins in the food industry.

## 1. Introduction

Anthocyanins are glycosylated derivatives of flavylium cations (2-phenylbenzopyrilium) with different methoxyl or hydroxyl substitutes on the rings, separated by an oxygen-containing six-membered heterocyclic ring [1,2]. They can be extracted from plants and have been broadly applied in the food industry as colourants, antioxidants, preservative materials, and functional additives [3,4]. However, anthocyanins are water-soluble natural pigments with low solubility in organic solvents and are vulnerable to external factors, such as high temperature, light, and strongly acidic and alkaline environments, which properties limit their applications in food matrices [5,6,7,8].

Acylated anthocyanins present potential alternatives to anthocyanins due to their unique functions and advantages compared with non-acylated analogues, attributed to the structural change in anthocyanins that is a result from the connection of acyl groups [9,10,11]. Acylated anthocyanins are anthocyanin derivatives with complex patterns of glycosylation and acylation (Figure 1). They exist naturally in many plant species and are distributed in virtually all plant parts, such as roots, stems, flowers, and fruits [6]. They can also be formed by chemical and enzymatic acylation in vitro and when so formed possess better stability and higher solubility in organic solvents when compared with non-acylated anthocyanins. Yang et al. [11] performed enzymatic acylation of blackcurrant anthocyanins and compared the thermal stability of the acylated anthocyanins with their parental anthocyanins. The results showed that the acylated anthocyanins had higher half-lives and activation energies compared to their parental anthocyanins, which indicated that the thermal stability of the anthocyanins was improved after acylation. Yan et al. [12] investigated the enzymatic acylation of black rice anthocyanins using methyl aromate as an acyl donor and showed that the acylation enhanced the thermal and light resistance of the anthocyanins. Currently, the commonly used methods for the structural identification of acylated anthocyanins include chromatographic methods, such as TLC, HPLC, and HPLC-MS [13,14], and spectroscopic methods, such as UV-Vis spectroscopy [15], FTIR [16], and NMR [17,18].

In order to better utilize acylated anthocyanins, there is a need for a comprehensive understanding of their formation from the perspectives of biosynthesis and chemical synthesis. Fortunately, with the continuous development of genetic engineering, molecular technology, and analytical chemistry, it is possible to better understand acylated anthocyanins at the level of gene expression and chemical synthesis. Proper understanding of the underlying mechanisms of acylation can facilitate the production and application of acylated anthocyanins in vivo and in vitro.

There are a few studies reported on the acylation and applications of anthocyanins. Giusti and Wrolstad [19] reviewed different edible sources of acylated anthocyanins and their applications in various food systems, containing dairy system as well as model juices. Martin et al. [20] focused on the chemical structure of anthocyanins, colour variation, and the potential benefits after intake of anthocyanins. Yousuf et al. [21] introduced an effective encapsulation approach to improve the stability of anthocyanins, increasing their range of potential uses. In addition, the extraction, potential health benefits, characterization, and delivery of anthocyanins were also provided in this review. For the acylation of anthocyanins, Zhao et al. [22] summarized the chemical significance of anthocyanin glycosyl acylation and the impacts of acylation on the stability of acylated anthocyanins. However, despite these reviews, the mechanisms of acylation and their applications in the food industry have been rarely covered. Therefore, this review intends to summarize the mechanisms of acylation as well as the application of acylated anthocyanins in the food industry and presents an outlook and potential prospect towards future research developments in the area.

## 2. Mechanisms of Anthocyanins Acylation

### 2.1. Biosynthetic Acylation of Anthocyanins

Acylation is the esterification of hydroxyl groups by aliphatic acyl donors or aromatic acyl donors and constitutes of a universally observed modification of plant secondary metabolites, contributing to various products with changeable physical and biological properties [23]. Acylated anthocyanins can be synthesized in vivo through the biosynthetic approach, beginning with phenylalanine, which is a main branch of the universal phenylpropanoid pathway [24]. As shown in Figure 2, the biosynthetic process from phenylalanine to anthocyanins is divided into three stages, including the general phenylalanine pathway, the early step of the flavonoid pathway and the last step of the specific anthocyanin pathway [25]. The biosynthesis of acylated anthocyanins is generally based on the synthesis of three common anthocyanins—cyanidin, pelargonidin and delphinidin—through glycosylation and acylation modifications with the corresponding participation of glycosyltransferase (GT) and acyltransferase (AAT) [26,27].

Due to the unique role of acyltransferase in the acylation of anthocyanins, research is geared towards identifying the key genes encoding acyltransferases for a proper understanding of the entire acylation process, and several key genes that encode acyltransferases in some plants have been identified with the help of genetic engineering and molecular biology. For the acylation via the biosynthetic approach in Arabidopsis, Luo et al. [28] incorporated co-expression profiling with anthocyanin accumulation and used the RT-PCR method to determine the expression levels of Arabidopsis thaliana BAHD genes in the anthocyanin acyltransferase super-clade. Their results indicated that 5 out of 10 genes were significantly upregulated (>4 times) under both stresses, and three genes (At3g29590, At1g03940, and At1g03495) closely associated with the production of three BAHD kinds of anthocyanin acyltransferases (AATs) were especially identified. The AATs utilize malonyl-CoA or p-coumaroyl-CoA as acting substrates to transfer three acyl groups of malonyl, sinapoyl, and p-coumaroyl to anthocyanin structures, which produce various acylated anthocyanins. In another study, Miyahara et al. [29] demonstrated that acyl-glucose-dependent glucosyltransferase (AAGT) in Arabidopsis played a key role in catalyzing the final step in the formation of anthocyanins, and reported AtBGLU10 as the key gene responsible for encoding the AAGT.

Several other studies have also been reported regarding key genes to encode acyltransferase in other plant tissues. In purple carrot, galactosyl was transferred from UDP-galactose to cyanidin by cyanidin galactosyltransferase (UCGalT), which enabled the anthocyanins to undergo glycosylation and acylation to form stable acylated anthocyanins. The acyltransferase involved in the acylation was produced by the expression of the DcSAT1 gene [30,31]. Moreover, Paulsmeyer et al. [32] reported one anthocyanin acyltransferase1 (AAT1) in maize, which was the first anthocyanin acyltransferase to be identified in a monocot species. The expression of AAT1 is related to the reduced acylation phenotype in maize, which causes a change in the proportion of acylated anthocyanins and anthocyanins in maize. Evidence also suggests that GRMZM2G387394 is a key gene to encode anthocyanin acyltransferase1 (AAT1) with anthocyanidin malonyltransferase activity in maize.

In berry, Sun et al. [33] conducted a transcriptomic analysis of black and white thorn grapes (*Vitis davidii*) to compare the difference in their transcriptional levels and look for several key genes associated with anthocyanin accumulation in *Vitis davidii*. They inferred from the results of the transcriptomic analysis that the gene 3AT (VIT-03s0017g00870) played a crucial role in the anthocyanin acylation process. Furthermore, based on the fact that anthocyanin acyltransferases belong to the BAHD family of proteins, Rinaldo et al. [34] established a unique accumulation of delphinidin-type anthocyanins in flower petals by means of heterologous expression of a flavonoid 30, 50-hydroxylaes gene to acquire bluish roses and carnations. The realization of this technology offers a latent breakthrough in the genetic engineering of different-coloured flowers.

In a recent study, tobacco lines were successfully developed by modifying key genes in acylated anthocyanin biosynthesis (Figure 2), thereby achieving high-level production of acylated anthocyanins [24]. This method for acylated anthocyanin production could be translatable to other plant species, such as Arabidopsis thaliana, which demonstrated the potential of this method via the production of a wide range of acylated anthocyanins, even though different plant species have different types of acylated anthocyanins with different synthesizing processes.

So far, a complete mechanism of acylation suitable for all kinds of plants is not available. However, the mechanism of acylation follows a general rule of synthesizing three common anthocyanins, followed by the modification of the anthocyanins via glycosylation and then acylation to obtain acylated anthocyanins under the control of glycosyltransferases and acyltransferases. Therefore, knowledge about the key regulatory genes for glycosyltransferases and acyltransferases can provide theoretical guidance for the commercial production of acylated anthocyanins.

### 2.2. Semi-Biosynthetic Acylation of Anthocyanins

The process of obtaining acylated anthocyanins via biosynthesis is a cumbersome method that involves a series of operations to identify and express the key regulatory genes. To solve this problem, the semi-biosynthetic method, which combines biosynthesis and chemical synthesis, has been proposed. The mechanism of the acylation of anthocyanins through semi-biosynthesis is mainly employed to synthesize non-acylated anthocyanins in plant cells through biosynthesis and then use acyl donors from the external environment to further synthesize specific acylated anthocyanins. The plant cell suspension culture process is the most commonly used semi-biosynthesis method for obtaining acylated anthocyanins [35]. According to this method, acyl donors are added to the cell culture medium in addition to the necessary substances for cell culture and the plant cells then use these substances to obtain target products during the growth process. This method is practicable and relatively simple compared with the process of biosynthesis. A summary of a number of studies on semi-biosynthetic anthocyanins is presented in Table 1.

Whittemore et al. [36] obtained five semi-biosynthetic acylated anthocyanins from Daucus carota (wild carrot) tissue cultures to investigate the mechanisms of colour loss via molecular modelling and analysis of stereo-electronic effects. Acylated anthocyanins synthesized via semi-biosynthesis in carrot culture are often mono-acylated and can be used to explore different properties of the same series of derivatives.

Apart from the choice of medium, the effects of certain external environmental factors, such as light, can also have important impacts on cell suspension culture. Andi et al. [37] compared the production of resveratrol, total phenols, and total flavonoids, including anthocyanins, in cell suspension culture under conditions of dark and light at a radiation of 135.1 μmols^−1^m^−2^ and with different concentrations of phenylalanine and methyl jasmonate. Their results showed that high yields of total phenols and total flavonoids could be obtained under the dark condition with a concentration of 1 mM phenylalanine and a concentration of 25 μM methyl jasmonate, indicating that during the plant cell suspension culture process, external environmental factors had a strong impact on the production and accumulation of flavonoids, such as anthocyanins and polyphenols, in plants.

Moreover, since obtaining acylated anthocyanins by the biosynthetic pathway usually involves at least eleven genes, as described in previous reports [27], it is relatively easier to obtain acylated anthocyanins by semi-synthesis rather than biosynthesis. Nevertheless, the semi-biosynthetic approach by plant cell culture involves the choice of a suitable plant cell system and requires a relatively harsh environment, which presents challenges in the use of this approach for acylated anthocyanin production. It is therefore important to select suitable plant cells for the acquisition of acylated anthocyanins, apart from optimizing the nutrients used in the medium, which is also an important factor affecting semi-synthesis. Furthermore, the key genes for acylated anthocyanins synthesis in plant cells can also be regulated by genetic engineering to formulate suitable plant cells necessary to obtain acylated anthocyanins through semi-synthesis. However, some genes of anthocyanin plants may change after gene modification, which may lead to some unexpected changes in phenotype, limiting the probability of obtaining high-yield plants by genetic engineering. In addition, the acquisition of acylated anthocyanins in nature is attended with difficulties in separation and extraction which are not conducive to industrial production.

### 2.3. Chemical Acylation of Anthocyanins

Acylation of anthocyanins can also be performed in vitro through chemical approaches. The key to chemical acylation is the formation of an ester group since there are lots of -OH groups in anthocyanins, including those of anthocyanidins and glycosyl groups. Therefore, in chemical approaches, anthocyanin acylation is divided into the acylation of anthocyanidins and the acylation of anthocyanin glycosyls.

Generally, the acylation of anthocyanins in plants mainly involves the partial or total esterification of -OH groups of anthocyanin glycosyl by various organic acids [22]. As with most chemical reactions, the species of catalyst, acylation sites, and the type and number of acyl groups all affect the chemical acylation [6,38]. When acyl donors are attached to hydroxyl groups, they can be folded and rotated, affecting the intramolecular co-pigmentation [22,39]. Thus, chemical acylation can significantly affect the stability of anthocyanins.

Recent studies related to the chemical acylation of anthocyanins are summarized in Table 1. In order to improve the stability of anthocyanins and broaden their applications in the food industry, Zhao et al. [40] synthesized acylated anthocyanins in blueberry by attaching lauric acid as the acyl group to the primary hydroxyl groups of cyanidin-3-glucoside during chemical acylation (Figure 3a). Since the ester group has higher stability than the hydroxyl group, the stability of the acylated cyanidin-3-glucoside is significantly higher than the non-acylated form. Apart from improving stability, chemical acylation also plays an important role in enhancing lipotropism. Anthocyanins are both hydrophilic and hydrophobic due to the presence of numerous hydrophilic hydroxyl groups, including phenolics and alcoholic hydroxyls, and a hydrophobic benzene ring in flavonoid compounds, but they usually show low lipotropism as a result of the existence of a great number of hydroxyls [41,42].

In order to obtain acylated anthocyanins with high lipotropism, Cruz et al. [43] utilized malvidin-3-O-glucoside separated from Porto red wine as a reactant during chemical acylation. This research showed for the first time the chemical synthesis of anthocyanins with stearoyl chloride in anhydrous acetonitrile overnight at room temperature and acquired malvidin-3-glucoside stearic acid conjugates with higher solubility in organic solvents (Figure 3b). Grajeda-Iglesias et al. [47] demonstrated that some polyphenols separated from vegetal extracts could be transformed into lipophilic colourants and obtained octanoyl derivatives of cyanidin-3-O-sambubioside and delphinidin-3-O-sambubioside via chemical acylation. Many sugars and organic acid moieties have also been used as acyl groups to obtain acylated anthocyanins during chemical acylation [43,48,49].

Although chemical acylation is a feasible approach to obtain acylated anthocyanins, the reactions are not region-selective, which may lead to unwanted functionalization of phenolic hydroxyl groups. External environmental factors, such as pH and ion concentration, affect the form and network of flavylium cations, hemiketals, quinonoid bases, and chalcones, which can lead to the transformation of anthocyanin structures and limit the progress of chemical acylation [39,43].

### 2.4. Enzymatic Acylation of Anthocyanins

Enzymatic acylation of anthocyanins displays higher region-selectivity with high yield compared with chemical acylation and can be used to synthesize some special acylated anthocyanins with high stability and fat solubility under special conditions [48]. Enzymatic acylation requires enzymes to participate in the acylation of anthocyanins; the essence of enzymatic acylation is similar to chemical acylation, which can also be regarded as a reaction in which hydroxyl groups are esterified by acyl donors, such as organic acids, carboxylic acid esters, and acid anhydrides, etc. The enzymes utilized in enzymatic acylation generally act in the immobilized form and may include proteases, lipases, and acyltransferases. Immobilization increases the stability of enzymes and improves the accessibility of substrates to the catalytic sites of enzymes, which helps to accelerate reactions [49]. The origin and concentration of the enzymes, the operating environment, and the structures of substrates are some of the factors that may affect enzymatic acylation [50,51]. Studies related to the enzymatic acylation of anthocyanins are summarized in Table 1.

**Table 1 foods-11-02166-t001:** Studies on the synthesis of acylated anthocyanins.

Methods	Substrate	Operating Conditions	Result	Reference
Semi-biosynthesis	Carrot cell culture; acyl donors: cinnamic and benzoic acid analogues	Concentrations of acids in Me_2_SO added to cultures at days 4 and 8 at a rate of 0.01 *v*/*v*	14 novel monoacylated anthocyanins	[52]
	Growing *Daucus carota* (wild carrot) tissue cultures; acyl donors: selected carboxylic acids		6-O-acyl-β-D-Glcp-(1 → 6)-β-D-Gal-(1 → O^3^)-cyanidin	[36]
	Lines of Del/Ros1/At3AT tobacco cells	Del/Ros1/At3AT tobacco cell, aromatic group	Cyanidin 3-O-(6″-O-(coumaroyl)glucoside)	[53]
	Tobacco suspension cultures	Nutrient medium (LS supplemented with 1 mg L^−^^1^ 2.4-D and 100 mg L^−^^1^ kanamycin)	Acylated cyanidin 3-O-(coumaroyl) rutinoside	[24]
Chemical acylation	Malvidin-3-glucoside; acyl donors: stearoyl chloride	Anhydrous acetonitrile solution, room temperature, argon atmosphere overnight	Mono- and di-ester derivatives of malvidin-3-glucoside	[43]
	Cyanidin-3-glucoside; acyl donors: lauric acid	DMF solution, 4 °C, argon atmosphere, 48 h	Acylatedcyanidin-3-glucoside	[40]
Enzymatic acylation	Malvidin-3-glucoside; acyl donors: oleic acid and linoleic acid; enzymes: lipase acrylic resin from *Candida antarctica* lipase B (CALB)	Anhydrous 2-methyl-2-butanol solution, stirred at 60 °C, argon atmosphere, 48 h	Malvidin-3-glucoside–oleic acid conjugate	[9]
	Cyanidin 3-glucoside; acyl donors: methyl benzoate, methyl salicylate; enzymes: Novozym 435	In pyridine solution, stirred at 40 °C, in a vacuum of 900 mbar, 48 h	Cyanidin-3-(6″-benzoyl)-glucoside, cyanidin-3-(6″-salicy-loyl)-glucoside, and cyanidin-3-(6″-cinnamoyl)-glucoside	[12]
	Malvidin 3-glucoside; acyl donors: fatty acids (from C4 to C16); enzymes: CALB	In dry 2-methyl-2-butanol solution, stirred, 60 °C, over 24 h	Malvidin-3-glucoside with fatty acid conjugates of different chain lengths	[44]
	Delphinidin-3-O-glucoside, cyanidin-3-O-glucoside; acyl donors: octanoic acid; enzymes: CALB	In dry acetonitrile–DMSO 10:1 (*v*/*v*) solution, stirred, 60 °C, 9 h	Delphinidin-3-glucoside-6″-O-octanoate, cyanidin-3-glucoside-6″-O-octanoate	[45]
	Cyanidin-3-glucoside; acyl donors: fatty acids (from C4 to C12); enzymes: CALB	In 2-methyl-2-butanol solution, stirred, 60 °C	Cyanidin-3-glucoside-fatty acid derivatives	[46]
	Cyanidin-3-O-galactoside; acyl donors: saturated fatty acids of different chain lengths; enzymes: (Novozyme 435)	In tertbutanol solution, stirred, 60 °C, 72 h	Cyanidin-3-O-(6″-dodecanoyl)-galactoside	[54]
	Delphinidin-3-O-glucoside, delphinidin-3-O-rutinoside, cyanidin-3-O-glucoside and cyanidin-3-O-rutinoside, acyl donors: lauric acid; enzymes: CALB	In tertbutanol solution, stirred, 60 °C, 72 h	Derivatives of Delphinidin-3-O-glucoside, delphinidin-3-O-rutinoside, cyanidin-3-O-glucoside and cyanidin-3-O-rutinoside	[11]

Cruz et al. [44] reported a 35% conversion rate of acylated anthocyanins when cyanidin-3-glucoside was acylated by saturated fatty acids with different carbon chain lengths via enzymatic acylation (Figure 3c). A lower conversion rate of 30.8% was observed by Zhao et al. [40] using similar substrates of cyanidin-3-glucoside and saturated fatty acids during chemical acylation. As biocatalysts, enzymes play important roles in accelerating reactions and increasing the conversion rate of anthocyanins during enzymatic acylation. It was reported in [46] that the conversion rates of different products decreased by increasing the length from C4 to C12 of the carbon chain of fatty acids during enzymatic acylation using cyanidin-3-glucoside and fatty acids (Figure 3d), confirming that the structure of the acyl donors can also affect the conversion rate.

In addition, when aromatic acids are utilized as acyl donors, the conversion rates of enzymatic acylation can vary greatly. Yan et al. [12] observed that the conversion rate could reach 91% when various aromatic acid methyl esters were used as acyl donors during lipase-catalyzed enzymatic acylation of anthocyanins from black rice under reduced pressures. The improved adaptability of aromatic acids in enzymatic acylation and the evaporation of the solvent at reduced pressure shifted the reaction equilibrium towards synthesis, leading to a relatively high conversion rate. The authors also verified the region-selectivity of enzymatic acylation reactions in the study, as three specific monoesters were successfully synthesized in the same position as a hydroxyl. Moreover, Luís Cruz et al. [45] isolated anthocyanins from blackcurrant skin waste (*Ribes nigrum* L.), which was acylated with octanoic acid under the catalysis of Candida Antarctica lipase B (Figure 3e) to obtain anthocyanin-3-glucoside-octanoate conjugates, namely, cyanidin-3-glucoside-6″-O-octanoate and delphinidin-3-glucoside-6″-O-octanoate. In another study, Cruz et al. [9] obtained malvidin-3-O-(6″-oleoyl) glucoside with higher lipotropism via enzymatic synthesis under the catalysis of lipase acrylic resin from Candida Antarctica, as shown in Figure 3f. In addition, de Castro et al. [50] synthesized monoesters from the primary hydroxyl group present on the C6 carbon atom glycoside moiety of the molecule (6″, -OH), which enabled region-selectivity since the glycosylated molecule was located towards the catalytic residues when the enzyme came into contact with the substrates.

Since anthocyanins are water-soluble natural pigments with low solubility in organic solvents, their strong hydrophilicity limits their application in lipophilic food matrices. Enzymatic acylation can produce acylated anthocyanins with improved fat solubility due to the participation of enzymes, presenting an additional method for the synthesis of acylated anthocyanins [11,44]. Therefore, acylated anthocyanins with better lipid solubility can be used in lipophilic food matrices, which extends the range of applications of anthocyanins in the food industry. However, in some enzymatic acylations, when fatty acids are used as acyl group donors, the conversion rate is relatively low. Therefore, future studies could focus on the influence of the structures and types of acyl donors on conversion rates. For example, esters could be chosen as acyl donors and volatile by-products could be evaporated, thus shifting the reaction equilibrium towards synthesis to improve the yield.

## 3. Applications of Acylated Anthocyanins

Natural anthocyanins are mainly derived from dark red, purple, or blue vegetables and fruits, such as black goji berries, purple potatoes, red kale, grapes, mulberries, blackberries, blueberries, and beetroot [55], and are a safe food additive with antioxidant, anticancer, and metabolism-regulating biological activities. However, applications of natural anthocyanins are limited due to their low stability and poor lipotropism. Therefore, acylated anthocyanins with high stability under different conditions have been extensively investigated. Among them, the common edible acylated anthocyanins are mainly derived from radishes, red potatoes, red cabbages, black carrots, and purple sweet potatoes [19]. Table 2 summarises the applications of acylated anthocyanins in the food industry in recent years, including colourants, functional agents, and indicator sensors for the food industry [8,56,57,58,59].

### 3.1. Food Colourants

The colour of food is an important organoleptic attribute that directly affects consumer choice. Given the safety issues associated with the use of synthetic food colourants, such as lemon yellow, the use of natural pigments, such as anthocyanins and their acylated forms, as food colourants is becoming more and more accepted due to their safe edibility and healthy properties [4,20], and a number of studies related to the discovery and application of anthocyanins and their acylated forms as food colourants have been published [62,78,79].

Oliveira et al. [64] isolated acylated anthocyanins, namely, peonidin-3-(6′-hydroxybenzoyl)-sophoroside-5-glucoside and peonidin-3-(6′-hydroxybenzoyl-6″-caffeoyl)-sophoroside-5-glucoside, from purple-fleshed sweet potatoes and showed high resistance to pH variations compared with the parent anthocyanins, demonstrating improved capability in maintaining red and blue colourants at acidic and basic pH. Similarly, Mendoza et al. [80] reported that three acylated units of sugar at the third position of the peonidin chromophore of heavenly blue anthocyanin were important to the stability of its purple and blue colours. The sugar units produced effective protective environments around flavylium cations through intramolecular sandwich-type stacking, which facilitated the stability of the colours of acylated anthocyanins. This finding could allow better use and transformation of different sources of paeoniflorin for effective and stable food colourants. Furthermore, Quan et al. [65] firstly identified the fractions of anthocyanins extracted from purple-fleshed sweet potatoes and subsequently evaluated the storage stability of anthocyanins for 6 months at 25 °C protected from light, 37 °C protected from light, and 25 °C under fluorescent conditions. The results showed that acylated anthocyanins, which contain diacylated and disaccharide groups of paeoniflorin derivatives, were more stable than other anthocyanins during storage, with much lower rates of anthocyanin degradation after 6 months of storage. Similarly, acylated anthocyanins extracted by Venter et al. [81] from plants of the Geraniaceae and Lamiaceae families remained stable over a wide temperature range, and these results suggest that Geraniaceae and Lamiaceae families have better potential applications as natural food colourants.

In addition, several food colourants have also been patented. Mane et al. [61] patented an anthocyanin-based colourant comprising acylated anthocyanins and pelargonidin-based anthocyanins with red colour having a specified hue value for uses in beverages, fruit preparations, dairy products, ice cream, and confectionary. In another patent submission, a natural colourant comprising acylated anthocyanins extracted from black goji berries exhibited colour stability for more than 14 days [63]. In the two patents, a natural food colourant rich in acylated anthocyanins together with a mixture of non-acylated anthocyanins were applied in a variety of food products, providing a reference for the development of natural colourants. The above-mentioned studies also described the current status and future applications of most commercial natural anthocyanins colourants, especially their effects on improving colour types, stability in food products, and their combination with other ingredients for enhancing customer experience.

### 3.2. Functionalizing Agents

Apart from being used as food colourants, epidemiological evidence associating anthocyanin intake with low incidences of chronic and degenerative diseases and the excellent antioxidant properties of acylated anthocyanins also suggest that they can be used as functionalizing agents to prevent cardiovascular illness, neuronal diseases, cancer, inflammation, and diabetes [1,82,83]. The human body produces a variety of free radicals in the process of metabolism, and excesses of free radicals lead to the oxidation of sugars, lipids, proteins, DNA, and RNA in the human body, which processes are related to cancer, Alzheimer’s disease, autoimmune deficiency disease, diabetes, etc. [15]. The reason why the intake of anthocyanins and acylated anthocyanins can reduce the incidence of chronic diseases is that they have excellent antioxidant properties and can have good free radical scavenging effects.

The antioxidant activity of anthocyanins is related to its purity and molecular structure. The simpler the molecular structure and the higher the purity, the stronger the antioxidant activity [84]. Moreover, the antioxidant activity of anthocyanin aglycone is higher than that of its glycosides [85], the antioxidant activity of monoglycosides is higher than that of polyglycones [82], and the antioxidant activity of anthocyanins is higher than that of acylated anthocyanins [85]. This is mainly because the phenolic hydroxyl group in the anthocyanin structure plays an important role in the scavenging mechanism of reactive oxygen radicals. The 3’, 4’ catechol structure on the B ring of anthocyanin can form a stable conjugated semiquinone or orthoquinone structure with RO through two consecutive single electron transfer reactions, which can evenly distribute the electron cloud and reduce intramolecular energy. The 5-position phenolic hydroxyl group of the A ring is easily oxidized to release H+ and has a strong ability to capture RO. The 3-, 5-, and 7-position phenolic hydroxyl groups further combine with RO to form a pseudo-semiquinone structure, and stability is improved through the keto–enol tautomerization reaction [83]. It has been reported that each ortho-substituted diphenol group can scavenge 4 mol RO [86].

In addition, the antioxidant activity of anthocyanins during storage is influenced by a variety of factors, such as light, temperature, pH, and storage status [87,88]. In particular, the storage state of anthocyanins, such as liquid and solid, has a significant difference on their antioxidant activities in different pH environments. When stored under acidic conditions (pH 1, 2, and 3), the antioxidant activity of liquid anthocyanin extracts was reduced less than that of solid anthocyanin extracts at the same pH conditions [89]. Relevant studies have explored the therapeutic effects of acylated anthocyanins on the basis of animal experiments. Hyperuricemia, a common disease, is caused by purine metabolism disorder and results in the accumulation of uric acid crystals in and around the human joints, leading to gout. Allopurinol therapy is a common treatment for hyperuricemia, but studies have associated allopurinol use with some adverse effects in patients [90,91,92,93,94]. It has also been reported that receiving allopurinol treatment causes serious drug-related adverse reactions in some patients [92,93]. Therefore, the therapeutic effects of acylated anthocyanins have been studied.

Zhang et al. [58] evaluated the therapeutic effects of acylated anthocyanins acquired from purple sweet potato along with allopurinol against hyperuricemia and kidney inflammation in hyperuricemic mice. Compared with the use of allopurinol alone, the combination of acylated anthocyanins and allopurinol was more effective in reducing serum uric acid levels and kidney damage in hyperuricemia mice. This study could present some new ideas in complementary therapy for hyperuricemia treatment and could lead to the application of anthocyanin-rich foods as health products.

Animal studies have also shown that acylated anthocyanins have great potential in trauma repair. Tsutsumi et al. [67] reported that acylated anthocyanins derived from purple carrot increased peripheral blood flow in the cremaster artery when administered orally to rats, implying that acylated anthocyanins in healthy foods could accelerate wound healing after surgery. Moreover, the consumption of natural foods rich in acylated anthocyanins, such as purple potatoes, was reported to affect postprandial inflammation, alleviate postprandial blood glucose levels and insulinemia, and decrease the risk of obesity [94]. With such encouraging results in animal studies, research efforts should be geared towards clinical studies for humans and the introduction of acylated anthocyanins in medicines or medical diets.

### 3.3. Indicators in Intelligent Packaging

Food packaging is an indispensable part of the food supply chain, aiming at ensuring the quality, safety, integrity, and wholesomeness of food products [95]. Traditional packaging can protect the quality and safety of food products but does not reflect freshness and quality in real-time. Therefore, intelligent packaging has been developed to ensure food quality and safety.

As the colour of anthocyanins and their acylated forms depends largely on the nature and structure of anthocyanins, which is mainly influenced by the external pH environment [39], anthocyanins and their acylated forms have been studied as pH-sensitive dyes for use in food packaging. As shown in Figure 4, the form of AH^+^ is usually red, the different forms of A are generally purple, and the forms of A^−^ are often blue. Since food spoilage is typically accompanied by a variation in pH, the quality and safety of packaged food products can be monitored by changes in the colours of sensitive anthocyanin dyes [96].

Choi et al. [73] developed an intelligent pH indicator film by fixing anthocyanin extract from purple sweet potato on an agar–starch support and used it to monitor pH change in pork samples by observing the colours at different pH values as reflections of spoilage. The colour of the film varying from red to green could reflect the deterioration of the pork with pH changes as shown in Figure 5. For seafood products, Eskandarabadi et al. [97] fabricated a novel pH-sensitive indicator that integrated black carrot anthocyanins, including acylated anthocyanins and bacterial nanocellulose, to monitor the freshness/spoilage of common crap fillet and rainbow trout during storage at 4 °C. The pH indicator showed distinguishable colours of deep carmine, charm pink, and jelly-bean blue corresponding to the quality levels of fresh, best to eat, and spoiled, respectively. Similarly, intelligent packaging that utilizes anthocyanin or acylated anthocyanins as a sensitive dye to monitor the quality and freshness of food products has also been developed for seafood [74,76], pasteurized milk [59], and shrimp [77].

Although the above studies show the feasibility of using anthocyanins and acylated anthocyanins as intelligent packaging materials, it is important to determine appropriate sensitive pigments and solid matrixes for packaging. The toxicity and edibility of these pigments should also be considered in the development of packaging.

## 4. Challenges and Future Trends

Although acylated anthocyanins have shown potential for applications in food colouring, functionalization, and intelligent packaging in the food industry due to their improved stability and enhanced fat solubility, there are still challenges and limitations in how to obtain acylated anthocyanins more efficiently and how to apply acylated anthocyanins more widely in the food industry. In the light of the available studies, this review provides some recommendations to address some of the limitations faced by acylated anthocyanins and a brief outlook on the future trends of their development.

(1)It is difficult to find suitable plants rich in acylated anthocyanins, and the direct isolation and purification of single acylated anthocyanins from a large variety of anthocyanins in plants is a relatively cumbersome and costly process. Therefore, low valueable food by-products could be used as good sources for the isolation of anthocyanins by countercurrent chromatography (HSCCC) in subsequent studies, which would solve the problem of obtaining raw materials as well as achieve multiple effective uses of resources. In addition, later studies could develop alternative techniques, such as gene editing, gene transfer, and heterologous expression techniques, to construct several acylated anthocyanin bio-factories to obtain highly abundant acylated anthocyanins.(2)In vitro synthesis of acylated anthocyanins faces challenges of low reaction yields; therefore, subsequent research on the semi-biosynthetic acylation of anthocyanins could focus on investigating more suitable types of plant cells and optimizing the composition of media in order to obtain high yields of acylated anthocyanins. In addition, for the synthesis of acylated anthocyanins using chemical acylation or enzymatic acylation, research into the best reaction conditions, e.g., by optimizing the solvents used in reaction systems, and ways of increasing enzyme activity and developing the hybrid synthesis of acylated anthocyanins is also important.(3)Acylated anthocyanins usually face the problem of insufficient colour variety when they are used as food colourants. In order to tackle this challenge, on the one hand, innovative food colourants with a wide pH range, which contain acylated anthocyanins and other types of natural dyes, could be developed. When using acylated anthocyanins as food colourants, adjusting the pH values of food products could also be a useful way of enriching the colour of food colourants. On the other hand, natural anthocyanin pigments/dyes and phenolic co-pigments/co-dyes could be used to form non-covalent complexes to enrich the variety of food colourants. Since these non-covalent interactions allow dyes to interact with each other to produce co-pigmentation, the colours in flowers, berries, and foods made from them (including wines, beverages, jams, purees, and syrups) could be stabilized and modulated.(4)Acylated anthocyanins have a variety of biological properties, such as anti-cancer, anti-inflammatory, anti-aging, anti-cardiovascular, and antioxidant properties, but the current studies on the biological activities of acylated anthocyanins are mostly at the cellular or preliminary mouse level. There are still not enough suitable primate models to validate the unique biological efficacies of acylated anthocyanins. Therefore, more animal models could be constructed, as well as relevant human clinical trials, to validate the biological activities of acylated anthocyanins in subsequent studies. In addition, safety evaluation models and methods for acylated anthocyanins need to be further refined.(5)Finally, when using acylated anthocyanins as indicators, a slight change in colour may not be observed by the naked eye, and thus their application in intelligent food packaging is challenging. More characterization methods could be introduced to reflect slight changes in colour. For example, traditional physical characteristics, such as CIELAB colour difference calculation, could be combined with molecular spectroscopy techniques, such as visible spectroscopy and near-infrared spectroscopy, to characterize changes in colour. New algorithms could also be developed to establish reliable correlation models to increase sensing accuracy.

## 5. Conclusions

The recent advances in the mechanisms of acylation, including biosynthesis, semi-biosynthesis, and chemical and enzymatic acylation, and their applications in the food industry have been summarized in the current review. The acylation of anthocyanins through biosynthesis is performed in vivo by gene editing and regulation, while semi-biosynthesis and chemical and enzymatic acylation of anthocyanins are performed in vitro through the intervention of the external environment. Although these methods have some difficulties in directly obtaining pure acylated anthocyanins, they can be used to obtain relatively large quantities of acylated anthocyanins. With the further development of technologies, such as gene identification, gene editing, and gene expression in vitro, it will be possible to systematically elaborate the mechanism of biosynthesis of acylated anthocyanins. In addition, in recent years, more and more attention has been paid to the derivatization of polyphenols, and many studies have been conducted on the selection of enzymes and the optimization of reaction systems, such that enzymatic acylation has become an efficient method for the derivatization of more natural active substances, leading to the enrichment of certain types of active substances, such as polyphenols, and the provision of more beneficial active substances.

## Figures and Tables

**Figure 1 foods-11-02166-f001:**
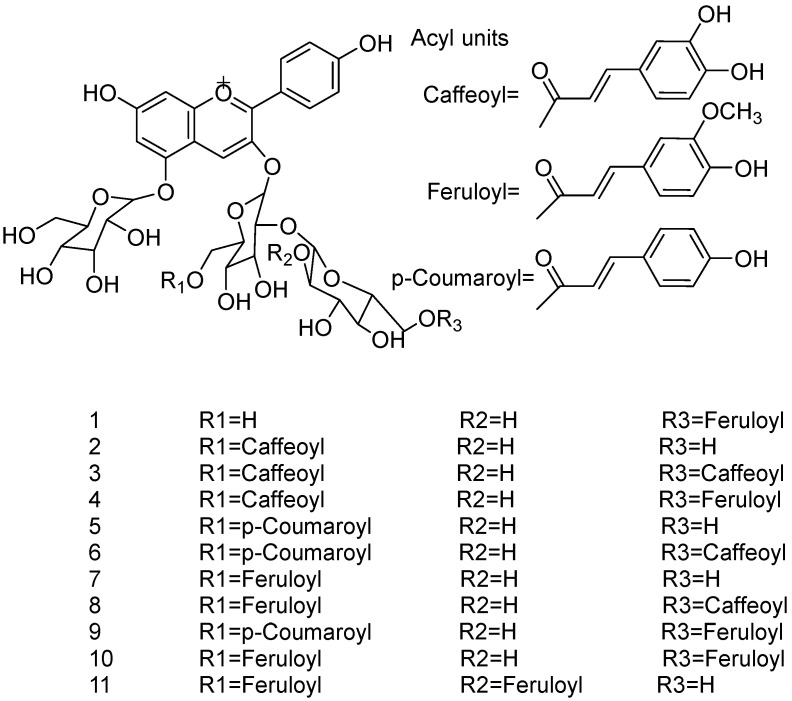
Structures of acylated anthocyanins from red radish extracts [6]. Reproduced with permission from Matsufuji, *J. Agric. Food Chem*.; published by ACS, 2007.

**Figure 2 foods-11-02166-f002:**
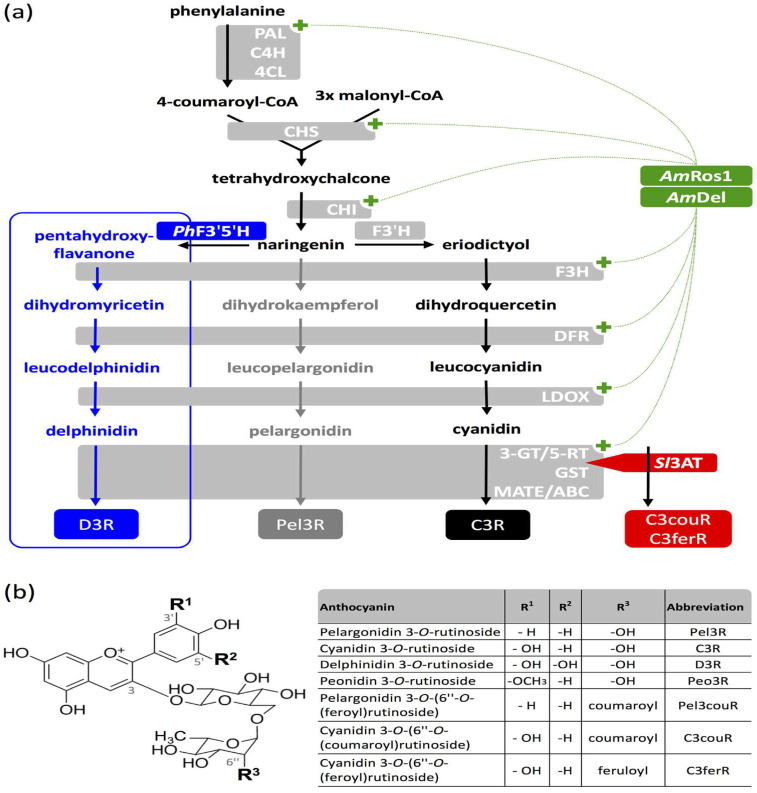
Engineered anthocyanin production in Nicotiana tabacum. (**a**) The biosynthesis pathway of anthocyanins. The expression of an anthocyanin 3-O-rutinoside-4′′′-hydroxycinnamoyl transferase from Solanum lycopersicum (Sl3AT) contributes to the production of aromatically acylated cyanidin 3-O-(6”-O-coumaroyl) rutinoside (C3couR) and cyanidin 3-O-(6′′-O-feruloyl) rutinoside (C3ferR). (**b**) Structures of anthocyanins obtained from tobacco cultures [24]. Reproduced with permission from Appelhagen, *Metab. Eng*.; published by Science Direct, 2018.

**Figure 3 foods-11-02166-f003:**
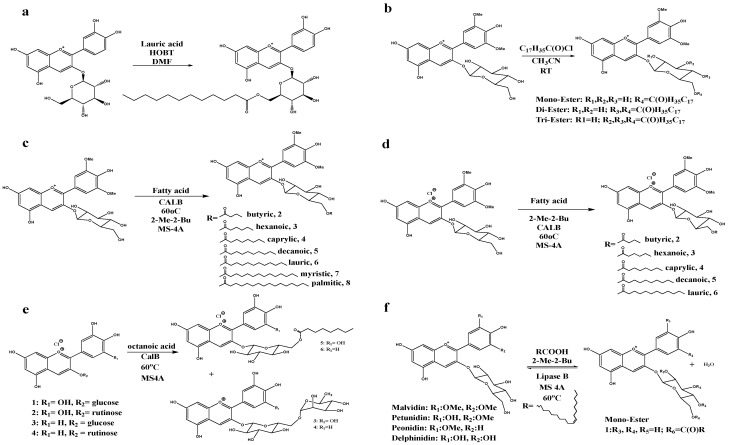
Examples of anthocyanin acylation. (**a**) The reaction scheme of derivatization of malvidin-3-glucoside with stearoyl chloride [43]. Reproduced with permission from Cruz, L, *Food Chem.*; published by Elsevier, 2015. (**b**) The chemical acylation between cyanidin-3-glucoside and lauric acid [40]. Reproduced with permission from Zhao, L.-y, *Int. J. Food Prop.*; published by Taylor and Francis Online, 2015. (**c**) The reaction of enzymatic esterification of the malvidin-3-glucoside [9]. Reproduced with permission from Cruz, L, *Food Funct.*; published by Croyal Society of Chemistry, 2016. (**d**) Enzymatic esterification reactions between malvidin 3-glucoside and different fatty acids [44]. Reproduced with permission from Cruz, L, *J. Agric. Food Chem.*; published by ACS, 2017. (**e**) Enzymatic esterification reactions between anthocyanins and octanoic acid under the catalysis of CALB [45]. Reproduced with permission from Cruz, L, *Food Chem.*; published by Elsevier, 2018. (**f**) Enzymatic esterification reactions between cyanidin-3-glucoside and various fatty acids [46]. Reproduced with permission from Grajeda-Iglesias, *Food Chem.*; published by Elsevier, 2017.

**Figure 4 foods-11-02166-f004:**
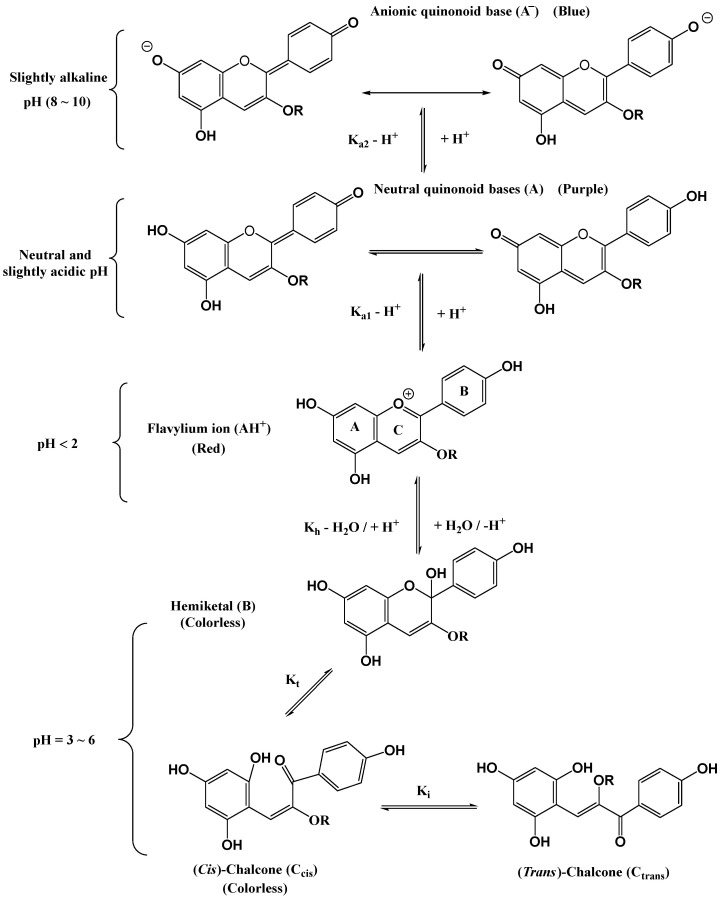
Structural transformations of anthocyanins in acidic to neutral solutions [39]. Reproduced with permission from Trouillas, P, *Chem. Rev.*; published by ACS, 2016.

**Figure 5 foods-11-02166-f005:**
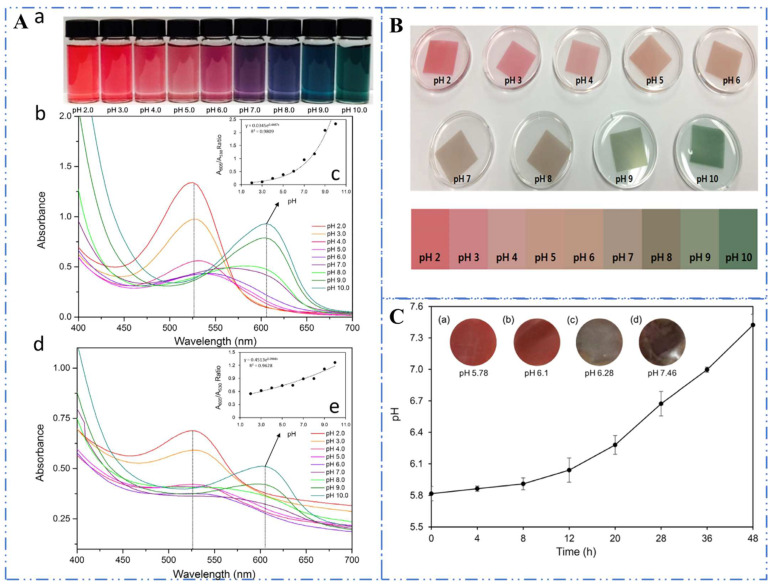
The scheme of a new colourimetric pH indicator film for the quality of pork. (**A**) Colour changes in PSPE solutions and intelligent films with PSPE. (**a**) Colour changes of PSPE solutions. (**b**) UV–vis spectra of PSPE solutions. (**c**) The absorbance ratio at 605 nm versus 530 nm of PSPE solutions. (**d**) UV–vis spectra of pH indicator films. (**e**) The absorbance proportion at 605 nm versus 530 nm of pH indicator films containing PSPE. (**B**) Colour changes in colourimetric intelligent pH indicator films dipped in various pH buffer solutions. (**C**) The curve of pH variations of pork samples and colour changes in intelligent pH indicator films of pork samples under accelerated storage conditions (25 °C) for various periods [73]. Reproduced with permission from Choi, I, *Food Chem.*; published by Elsevier, 2017.

**Table 2 foods-11-02166-t002:** Studies on the applications of acylated anthocyanins in the food industry.

Application Categories	Sources	Major Acylated Anthocyanins	Specific Functions or Application Fields	Reference
Food colourants	*Ajuga reptans* flowers and corresponding cell cultures	Delphinidin 3-(p-coumaroylferuloyl) sophoroside-5-malonylglucoside, delphinidin 3-(diferuloyl) sophoroside-5-malonylglucoside, and cyanidin 3-(di-p-coumaroyl) sophoroside-5-glucoside	Food colourants in the food industry	[60]
		Anthocyanins and pelargonidin-based anthocyanins	Beverages, fruit preparations, dairy products, ice cream, and confectionary	[61]
	*Thymus moroderi* and another five Thymus spp.	Cyanidin dimalonyldiglucoside, cyanidin 3-malonyl-glucoside, cyanidin 3-malonyl-acetylglucoside, peonidin dimalonyl-diglucoside, and pelargonidin 3-malonyl-acetyl-glucoside.	Food colourants in yoghurts	[62]
	Black goji berry (*Lycium ruthenicum Murr*.)	Acylated anthocyanins	Food colourants in food products	[63]
	Purple-fleshed sweet potato	Peonidin-3-(6′-hydroxybenzoyl)-sophoroside-5-glucoside, peonidin-3-(6′-hydroxybenzoyl-6″-caffeoyl)-sophoroside-5-glucoside.	A higher capability in retaining red and blue colours	[64]
	Purple-fleshed sweet potato	Peonidin 3-p-hydroxybenzoyl sophoroside-5-glucoside, peonidin 3-feruloyl sophoroside-5-glucoside, peonidin 3-caffeoyl sophoroside-5-glucoside, peonidin dicaffeoyl sophoroside-5-glucoside, peonidin 3-caffeoyl-p-hydroxybenzoyl sophoroside-5-glucoside and peonidin caffeoyl-feruloyl sophoroside-5-glucoside.	Natural colourant in food industry	[65]
	Sohiong (*Prunus nepalensis* L.)	Anthocyanins	Yoghurt, syrup, and hard-boiled candy	[66]
Functionalizing agents	Purple sweet potato (*Ipomoea batatas* L.)	Peonidin 3-(6′,6″-dicaffeoyl sophoroside)-5-glucoside, peonidin 3-(6′-caffeoyl-6″-p-hydroxybenzoyl sophoroside)-5-glucoside, peonidin 3-(6′-caffeoyl-6″-feruloyl sophoroside)-5glucoside	Alleviating hyperuricemia and kidney inflammation	[58]
	Purple sweet potato (*Ipomoea batatas* L. cultivar Eshu No.8)	Cyanidin-3-caffeoyl-feruloyl sophoroside-5-glucoside and peonidin-3-dicaffeoyl sophoroside-5-glucoside	Bioprotective activity and antioxidant capacity	[10]
	Purple carrot (*Daucus carota* L.)	Cyanidin-3-(2″-xylose-6″-sinapoyl-glucose-galactoside), cyanidin-3-(2″-xylose-6″-feruloyl-glucose-galactoside), cyanidin-3-(2″-xylose-6″(4-coumaroyl) glucose-galactoside)	Anthocyanins might possess adrenomimetic properties and be applied in wound recovery	[67]
	Purple root tubers and leaves of sweet potato (*Ipomoea batatas*)	Peonidin derivatives and cyanidin derivatives	Anti-proliferative activity	[68]
	Blackcurrant (*Ribes nigrum*)	Delphinidin-3-O-glucoside, delphinidin-3-O-rutinoside, cyanidin-3-O-glucoside and cyanidin-3-O-rutinoside derivatives	Antioxidant capacity and applied in inhibiting lipid peroxidation	[11]
	Alpine bearberry (*Arctostaphylos alpina*)	Cyanidin-3-O-(6″-dodecanoyl) galactoside	Antioxidant capacity applied in lipophilic food, cosmetic, and pharmaceutical products	[54]
	Purple sweet potato	Cyanidnin-(3-caffeylferulysophoroside-5-glucoside), peonidin-(3-caffeylferulysophoroside-5-glucoside)	Anti-obesity and antioxidative effects	[69]
	Red cabbage microgreens	Cyanidin(3-(glucosyl) (sinapoyl), (p-coumaroyl) sophorside-5-glucoside), cyanidin-(3-(glucosyl) (sinapoyl) (feruloyl)sophorside-5-glucoside)	Anti-obesity effect	[70]
Intelligent packaging	Grapes (*Christian Hansen*)	Anthocyanin extracted from grapes	Monitoring pH variations	[71]
	Red cabbage (*Brassica oleracea var capitata*)	Anthocyanins from red cabbage (*Brassica oleracea var capitata*)	Time–temperature indicators to detect pasteurized milk	[59]
	The flowers of rose and red cabbage	Anthocyanins from the flowers of rose and red cabbage	pH indicators to detect the freshness of buffalo meat	[72]
	Purple sweet potato	Purple sweet potato anthocyanins	pH indicators for the quality of pork	[73]
	Black rice	Black rice bran anthocyanins	Intelligent film for seafood spoilage monitoring	[74]
	Black carrot	Black carrot anthocyanins	pH-sensing indicator for monitoring fish freshness	[75]
	Purple sweet potato	Purple sweet potato anthocyanins	pH-sensitive films to monitor fish freshness	[76]
	Purple sweet potato and red cabbage	Anthocyanins from purple sweet potato and red cabbage	pH-sensitive films for the detection of shrimp deterioration	[77]

## Data Availability

Not applicable.

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
