# Peer review of "Acylation of Anthocyanins and Their Applications in the Food Industry: Mechanisms and Recent Research Advances"

_foods, 2022, doi:10.3390/foods11142166_

Round 1
Reviewer 1 Report
Dear Authors,
The review with title "Acylation of Anthocyanins and Their Applications in the Food 2 Industry: Mechanisms and Recent Research Advances" is very interesting and deals with very important overview of the mechanisms of acylation and the applications of acylated anthocyanins in the food industry. Acylated anthocyanins are very important natural pigment with strong antioxidant potential which can be excellent replacement of cancerogenic synthetic azo-pigments. However, I will suggest improvement of the review in few points by including :
- discussion of the most used techniques for structural determination of acylated anthocyanins (HPLC-MS, MS/MS fragmentation and NMR structural elucidation);
- low valuable food by-products which can be used as good source for isolation of anthocyanins by countercurrent chromatography (HSCCC);
-relationship between antioxidant potential and structural elucidation of anthocyanins (is acylation, mono and di-glycosidic bonding) enhance their antioxidant potential?
-effect of the storage on the antioxidant potential of acylated anthocyanins at different pH values.
Kind Regards
Reviewer 2 Report
This article sheds light on the recent advances in the mechanisms of acylation and also on food industrial applications of acylated anthocyanins. The review is well structured and touches on the crucial points of the above-mentioned subject. I feel that some major challenges regarding the application of different food products lack a more comprehensive discussion, and that would certainly improve the manuscript. Apart from this, below are some minors that I would like to see addressed by the authors.
Abstract. The authors must follow the publishing guidelines (structured abstract without headings; max number of words). I also advise reviewing the manuscript keywords (present in the title, a bit vague like "applications", "mechanisms").
L49. Please describe the "unique functions" that characterize the acylated anthocyanins.
L83. Please elaborate on what products display changeable physical and biological properties.
L292. Elaborate on the importance of the fat solubility feature.
L300. In my point of view, the manuscript would benefit if the natural sources of acylated anthocyanins are clearly pointed out.
L308. Can you give some (comprehensive) information on the stability evidenced by acylated pigments under different environmental conditions like pH?
L339. Table 2 is a very interesting one. Is it possible to better divide or link the purposes to the actual sources? This way seems kind of confusing.
L415-417. Please revise the sentence, it is a bit confusing.
L440. Do you mean in-silico models?
Reviewer 3 Report
Dear Editor,
The aim of this manuscript is to review current research on acylation of anthocyanins and their usage in food industry. I think such review is timely, need for natural food colourants is widely known. Especially I like chapter Challenges and future trends, it states clear problems and themes for further research. In general I like manuscript and its structure, but I have some minor suggestions:
Latin names must be in Italic
Table 1 and 2 are too large, maybe move to supplements?
Figure 5 meaning of ± must be given (SE, SEM, etc.) and confidence level.
Round 2
Reviewer 1 Report
Dear Authors,
The revised version of the manuscript with title "Acylation of Anthocyanins and Their Applications in the Food Industry: Mechanisms and Recent Research Advances" is significantly improved and all my questions and suggestions are accepted and well answered and discussed. I suggest acceptance of the manuscript in this revised version and publication in "Foods" as high quality journal.
Kind Regards